# DYNAMIC ROUTING MIXTURE OF EXPERTS FOR ENHANCED MULTI-LABEL IMAGE CLASSIFICATION

## ABSTRACT

Multi-label image classification (MLC) is a fundamental task in computer vision, requiring the identification of multiple objects or attributes within a single image. Traditional approaches often rely on shared backbones and static gating mechanisms, which can struggle to effectively capture complex label correlations and handle label heterogeneity, leading to issues such as negative transfer. In this paper, we introduce the **Dynamic Routing Mixture of Experts (DR-MoE)** model, a novel architecture that integrates input-dependent dynamic gating networks into the mixture-of-experts (MoE) framework for MLC. Unlike static gating in existing models like the Hybrid Sharing Query (HSQ) Yin et al. (2024), our dynamic gating mechanism adaptively selects and weights both shared and task-specific experts based on the input image features. This allows DR-MoE to better capture varying label dependencies and mitigate negative transfer, resulting in improved overall and per-label classification performance. We conduct extensive experiments on benchmark datasets MS-COCO Lin et al. (2014) and PASCAL VOC 2007 Everingham et al. (2015), demonstrating that DR-MoE achieves state-of-the-art results, outperforming existing methods including HSQ, Q2L Liu et al. (2021), and ML-GCN Chen et al. (2019). Additionally, ablation studies confirm the effectiveness of dynamic gating in enhancing model adaptability and performance, particularly for labels with high heterogeneity. Our findings suggest that incorporating dynamic routing mechanisms into MoE architectures is a promising direction for advancing multi-label image classification.

## 1 INTRODUCTION

Multi-label image classification (MLC) aims to assign multiple labels to an image, reflecting the presence of various objects or attributes within the scene. Unlike single-label classification, where each image is associated with a single category, MLC must handle the complexity of predicting a set of labels that may exhibit intricate relationships, including co-occurrence and mutual exclusivity. This task is fundamental in computer vision applications such as image tagging, scene understanding, medical diagnosis, and autonomous driving Wang et al. (2016); Chen et al. (2019).

Existing methods often employ shared backbones to extract features and capture label correlations through shared parameters Chen et al. (2019); Lanchantin et al. (2021). However, learning multiple labels jointly can lead to negative transfer, where optimizing for one label adversely affects the performance of others due to label heterogeneity. This problem is exacerbated when labels have conflicting features or when the model cannot adequately disentangle shared and label-specific information.

To address the negative transfer issue, the Hybrid Sharing Query (HSQ) model Yin et al. (2024) formulates MLC as a multi-task learning problem and introduces a mixture-of-experts (MoE) architecture with shared and task-specialized experts. HSQ leverages shared experts to capture common patterns among labels and task-specialized experts to handle label-specific features. A static gating mechanism combines the outputs of these experts, aiming to balance shared and unique representations.

However, the static gating in HSQ may not fully capture the dynamic nature of label correlations that vary across different images. In real-world scenarios, the relevance of shared and task-specific experts can change depending on the content of each image. For instance, an image containing

both *bicycle* and *person* may benefit from shared features, while an image with only *bicycle* may require more emphasis on task-specific features. Static gating cannot adapt to these variations, potentially limiting the model's ability to exploit positive label correlations and mitigate negative transfer effectively.

In this paper, we propose the **Dynamic Routing Mixture of Experts (DR-MoE)** model for enhanced multi-label image classification. Our approach introduces input-dependent dynamic gating networks that adaptively select and weight experts based on each individual image. By making the gating mechanism a function of the input features, DR-MoE allows the model to tailor the combination of shared and task-specific experts to the specific content of each image. This dynamic routing enables better capture of varying label dependencies and more effective handling of label heterogeneity.

Our contributions can be summarized as follows:

- We propose DR-MoE, a novel model that integrates dynamic gating networks into the MoE framework for multi-label image classification, enabling adaptive expert selection based on input images.

- We introduce input-dependent gating mechanisms that allow the model to capture dynamic label correlations and mitigate negative transfer by customizing expert utilization per sample.

- We demonstrate that DR-MoE achieves state-of-the-art performance on benchmark datasets such as MS-COCO Lin et al. (2014) and PASCAL VOC Everingham et al. (2015), outperforming existing methods including HSQ.

- We provide extensive experiments and analyses to show that dynamic routing improves per-label accuracy, especially for labels with high heterogeneity, and offers insights into the model's adaptive behavior.

## 2 RELATED WORK

Multi-label image classification (MLC) has been extensively studied in computer vision, with various approaches proposed to address the challenges of label correlations and heterogeneity. In this section, we review existing methods in MLC, the use of mixture-of-experts (MoE) architectures in deep learning, and dynamic routing mechanisms, highlighting the gap that our proposed model aims to fill.

### 2.1 MULTI-LABEL IMAGE CLASSIFICATION APPROACHES

Early approaches to MLC leveraged convolutional neural networks (CNNs) with shared backbones to extract image features, followed by classifiers for each label Wang et al. (2016; 2017). However, these methods often failed to capture complex label dependencies and struggled with label heterogeneity. To address label correlations, graph-based methods introduced graph convolutional networks (GCNs) to model label relationships explicitly Chen et al. (2019); Ye et al. (2020). For example, Chen *et al.* Chen et al. (2019) proposed ML-GCN, which utilizes GCNs to learn label embeddings and exploit inter-label correlations.

Transformer-based models have recently gained attention in MLC due to their ability to capture long-range dependencies. The Query2Label (Q2L) model Liu et al. (2021) employs a transformer decoder with learnable query embeddings representing each label. The model attends to the image features extracted by a CNN backbone and generates predictions for each label through the transformer mechanism. Similarly, Lanchantin *et al.* Lanchantin et al. (2021) proposed a general multi-label classification framework using transformers, demonstrating improved performance over CNN-based models.

While these methods effectively model label correlations, they often rely on shared parameters and may not adequately handle label heterogeneity. Learning all labels jointly can lead to negative transfer, where the optimization for some labels adversely affects others. This issue motivates the need for architectures that can disentangle shared and label-specific features.

## 2.2 MIXTURE-OF-EXPERTS IN DEEP LEARNING

Mixture-of-Experts (MoE) architectures have been employed to address the challenges of multi-task learning and to improve model capacity Jacobs et al. (1991); Shazeer et al. (2017). In MoE models, multiple experts specialize in different aspects of the data, and a gating mechanism determines how to combine their outputs. This allows the model to capture diverse patterns and allocate resources effectively.

In the context of MLC, the Hybrid Sharing Query (HSQ) model Yin et al. (2024) introduced an MoE architecture with shared and task-specialized experts to handle label correlations and heterogeneity. HSQ employs a static gating mechanism that combines the outputs of shared experts, which capture common patterns among labels, and task-specific experts, which focus on label-specific features. This approach aims to leverage positive label correlations while mitigating negative transfer.

However, the static gating in HSQ does not adapt to the varying label dependencies across different images. The fixed combination of experts may not fully exploit the potential of MoE architectures, especially when the relevance of shared and task-specific experts changes with the input.

## 2.3 DYNAMIC ROUTING MECHANISMS

Dynamic routing mechanisms have been proposed to enable models to adaptively select computational paths based on the input data Sabour et al. (2017). In capsule networks, dynamic routing allows capsules to route their outputs to appropriate higher-level capsules, capturing hierarchical relationships in the data.

In natural language processing, dynamic MoE models employ input-dependent gating networks to route tokens to experts Riquelme et al. (2021); Du et al. (2022). For instance, the GShard model Lepikhin et al. (2021) uses a dynamic routing algorithm to enable large-scale MoE models for machine translation, allowing the model to adaptively allocate experts to different tokens.

These dynamic routing mechanisms have shown promise in capturing input-dependent patterns and improving model efficiency. However, their application to MLC, particularly in handling label correlations and heterogeneity through adaptive expert selection, has not been fully explored.

## 2.4 GAP IN EXISTING RESEARCH

Despite the advances in MLC and MoE architectures, existing methods often lack input-dependent gating mechanisms that can adaptively handle varying label dependencies on a per-sample basis. Static gating in models like HSQ may not fully capture the dynamic nature of label correlations, potentially limiting their ability to exploit positive correlations and mitigate negative transfer.

Our proposed Dynamic Routing Mixture of Experts (DR-MoE) model addresses this gap by introducing input-dependent dynamic gating networks into the MoE framework for MLC. By allowing the gating mechanisms to consider the input features, DR-MoE can adaptively select and weight experts based on each individual image, enhancing the model's capacity to handle label correlations and heterogeneity effectively.

## 3 METHODOLOGY

In this section, we present the proposed **Dynamic Routing Mixture of Experts (DR-MoE)** model for multi-label image classification. Our model integrates input-dependent dynamic gating networks into the mixture-of-experts (MoE) framework, allowing adaptive expert selection based on each input image. The architecture consists of four main components: (1) a feature extraction backbone, (2) a transformer-based query module with learnable classification tokens, (3) shared and task-specific experts, and (4) dynamic gating networks that adaptively fuse expert outputs. Figure 1 illustrates the overall architecture of DR-MoE.

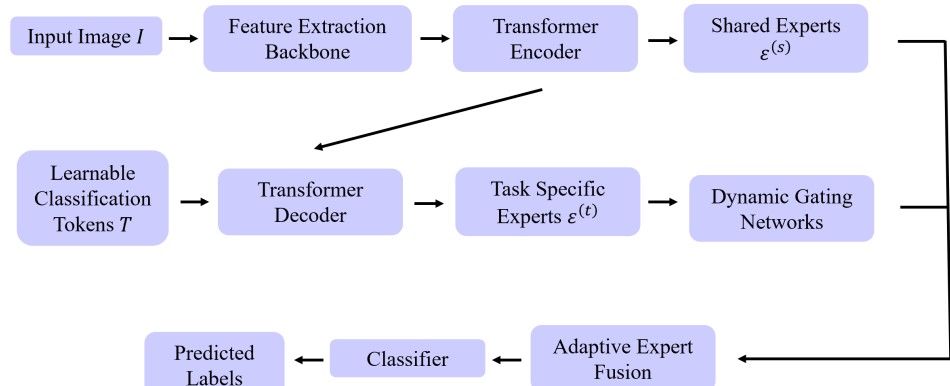

Figure 1: Overview of the proposed DR-MoE architecture for multi-label image classification. The model consists of a feature extraction backbone (e.g., ResNet-101), a transformer-based query module with learnable classification tokens for each label, shared and task-specific experts, and dynamic gating networks that adaptively combine expert outputs based on input features.

## 3.1 FEATURE EXTRACTION BACKBONE

We employ a convolutional neural network (CNN) backbone to extract feature representations from input images. Given an input image $\mathbf{I} \in \mathbb{R}^{3 \times H \times W}$, where $H$ and $W$ denote the height and width, respectively, the backbone produces a feature map $\mathbf{F} \in \mathbb{R}^{C \times H' \times W'}$, where $C$ is the number of channels, and $H'$, $W'$ are the spatial dimensions after downsampling.

We utilize pre-trained networks such as ResNet-101 He et al. (2016) or ConvNeXt Liu et al. (2022) as the backbone. The choice of backbone can be adapted based on computational resources and performance requirements. The extracted feature maps serve as input to the transformer-based query module.

## 3.2 TRANSFORMER-BASED QUERY MODULE

To capture label-specific features and model inter-label relationships, we employ a transformer-based query module inspired by prior works Vaswani et al. (2017); Liu et al. (2021). The module uses learnable classification tokens as queries, allowing the model to attend to relevant features for each label.

We introduce a set of $L$ learnable classification tokens $\mathbf{T} = [\mathbf{t}_1, \mathbf{t}_2, \ldots, \mathbf{t}_L]^\top \in \mathbb{R}^{L \times D}$, where $L$ is the number of labels and $D$ is the embedding dimension. Each token $\mathbf{t}_i$ corresponds to a specific label and is initialized randomly. These tokens are used to query the feature representations extracted by the backbone.

The flattened feature map $\mathbf{F}_{\text{flat}} \in \mathbb{R}^{(H'W') \times C}$ is passed through a transformer encoder to capture global context. Positional encodings are added to retain spatial information. The encoder consists of $N_{\text{enc}}$ layers of multi-head self-attention and feed-forward networks.

The self-attention mechanism is defined as:

$$\text{Attention}(\mathbf{Q}, \mathbf{K}, \mathbf{V}) = \text{Softmax}\left(\frac{\mathbf{Q}\mathbf{K}^\top}{\sqrt{d_k}}\right)\mathbf{V}, \tag{1}$$

where $\mathbf{Q}$, $\mathbf{K}$, $\mathbf{V}$ are the query, key, and value matrices, and $d_k$ is the dimension of the key vectors.

## 3.3 TRANSFORMER DECODER

The learnable classification tokens $\mathbf{T}$ serve as queries in the transformer decoder, which attends to the encoded features from the encoder. The decoder consists of $N_{\text{dec}}$ layers, each comprising multi-head self-attention, cross-attention, and feed-forward networks.

For each label $i$, the decoder outputs a label-specific representation:

$$\mathbf{h}_i = \text{Decoder}(\mathbf{t}_i, \mathbf{F}_{\text{enc}}), \tag{2}$$

where $\mathbf{F}_{\text{enc}}$ denotes the output of the transformer encoder.

## 3.4 SHARED AND TASK-SPECIFIC EXPERTS

To handle label correlations and heterogeneity, we employ a set of shared experts and task-specific experts. The shared experts capture common patterns across labels, while the task-specific experts focus on label-specific features.

Let $\mathcal{E}^{(s)} = \{E_1^{(s)}, E_2^{(s)}, \ldots, E_{N_s}^{(s)}\}$ denote the set of $N_s$ shared experts, and $\mathcal{E}^{(t)} = \{E_1^{(t)}, E_2^{(t)}, \ldots, E_L^{(t)}\}$ denote the set of $L$ task-specific experts, one for each label. Each expert is implemented as a feed-forward network (FFN) with ReLU activations:

$$E_j^{(s)}(\mathbf{x}) = \text{ReLU}(\mathbf{x}\mathbf{W}_j^{(s)} + \mathbf{b}_j^{(s)}), \tag{3}$$

$$E_i^{(t)}(\mathbf{x}) = \text{ReLU}(\mathbf{x}\mathbf{W}_i^{(t)} + \mathbf{b}_i^{(t)}), \tag{4}$$

where $\mathbf{W}_j^{(s)}, \mathbf{b}_j^{(s)}, \mathbf{W}_i^{(t)}$, and $\mathbf{b}_i^{(t)}$ are the weights and biases of the experts.

For each label $i$, the shared experts process the label-specific representation $\mathbf{h}_i$ to produce shared expert outputs:

$$\mathbf{s}_{i,j} = E_j^{(s)}(\mathbf{h}_i), \quad \text{for } j = 1, 2, \ldots, N_s. \tag{5}$$

The task-specific expert for label $i$ processes $\mathbf{h}_i$ to produce a task-specific output:

$$\mathbf{t}_i = E_i^{(t)}(\mathbf{h}_i). \tag{6}$$

## 3.5 DYNAMIC GATING NETWORKS

The dynamic gating networks generate input-dependent gating weights that adaptively combine the outputs of shared and task-specific experts for each label.

For each label $i$, we define a gating network $G_i$ implemented as a multi-layer perceptron (MLP) with ReLU activations and a softmax output layer:

$$\mathbf{w}_i = \text{Softmax}(G_i(\mathbf{h}_i)) \in \mathbb{R}^{N_s+1}, \tag{7}$$

where $\mathbf{w}_i = [w_{i,1}^{(s)}, \ldots, w_{i,N_s}^{(s)}, w_i^{(t)}]^\top$ contains the gating weights for the shared experts and the task-specific expert.

The gating network $G_i$ takes the label-specific representation $\mathbf{h}_i$ as input and outputs a probability distribution over the experts.

## 3.6 ADAPTIVE EXPERT FUSION

The final output for label $i$ is a weighted sum of the expert outputs:

$$\mathbf{o}_i = \sum_{j=1}^{N_s} w_{i,j}^{(s)} \mathbf{s}_{i,j} + w_i^{(t)} \mathbf{t}_i. \tag{8}$$

This adaptive fusion allows the model to focus on the most relevant experts for each input image and label, capturing dynamic label dependencies and mitigating negative transfer.

## 3.7 CLASSIFICATION AND LOSS FUNCTION

The final logits for label $i$ are obtained by applying a linear classifier to $\mathbf{o}_i$:

$$\hat{y}_i = \mathbf{W}_c^\top \mathbf{o}_i + b_c, \tag{9}$$

where $\mathbf{W}_c$ and $b_c$ are the weights and bias of the classifier.

We use the binary cross-entropy loss for multi-label classification:

$$L = \frac{1}{L} \sum_{i=1}^{L} \left[ -y_i \log(\sigma(\hat{y}_i)) - (1 - y_i) \log(1 - \sigma(\hat{y}_i)) \right], \tag{10}$$

where $y_i \in \{0, 1\}$ is the ground-truth label, and $\sigma(\cdot)$ denotes the sigmoid function.

## 3.8 TRAINING PROCEDURE

The entire model, including the backbone, transformer module, experts, and gating networks, is trained end-to-end using the Adam optimizer Kingma & Ba (2014) with weight decay for regularization. We apply data augmentation techniques such as random cropping, flipping, and color jittering during training.

## 3.9 REGULARIZATION TECHNIQUES

To prevent the gating networks from becoming overly confident or sparse too quickly, we apply entropy regularization to the gating weights:

$$R_{\text{entropy}} = -\lambda_{\text{entropy}} \frac{1}{L} \sum_{i=1}^{L} \sum_{k=1}^{N_s+1} w_{i,k} \log w_{i,k}, \tag{11}$$

where $\lambda_{\text{entropy}}$ is a hyperparameter controlling the strength of the regularization.

The total loss becomes:

$$L_{\text{total}} = L + R_{\text{entropy}} + R_{\text{weight\_decay}}, \tag{12}$$

where $R_{\text{weight\_decay}}$ is the weight decay regularization term.

## 3.10 VISUALIZATION OF DYNAMIC ROUTING

To illustrate the adaptive behavior of the dynamic gating networks, we visualize the gating weights for different input images. Figure 2 shows examples where the model assigns higher weights to shared experts when labels are correlated and higher weights to task-specific experts when labels are heterogeneous.

## 3.11 IMPLEMENTATION DETAILS

For reproducibility, we provide key implementation details:

- **Backbone:** We use ResNet-101 He et al. (2016) pre-trained on ImageNet Deng et al. (2009) as the feature extraction backbone.
- **Transformer Parameters:** The transformer encoder and decoder each have $N_{\text{enc}} = N_{\text{dec}} = 2$ layers, with embedding dimension $D = 512$ and 8 attention heads.
- **Experts:** We use $N_s = 4$ shared experts and $L$ task-specific experts. Each expert is an FFN with hidden dimension 256.
- **Gating Networks:** Each gating network is an MLP with one hidden layer of size 256, ReLU activation, and a softmax output layer.
- **Training:** We train the model for 50 epochs with batch size 32, using the Adam optimizer with learning rate $1 \times 10^{-4}$ and weight decay $1 \times 10^{-5}$.
- **Regularization:** We set $\lambda_{\text{entropy}} = 0.01$ for entropy regularization.
- **Data Augmentation:** Random resized cropping, horizontal flipping, and color jitter are applied during training.

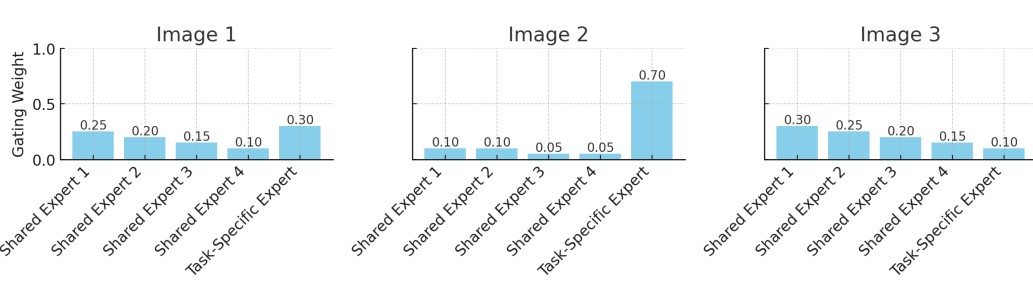

Figure 2: Visualization of dynamic gating weights for different input images. The model adaptively adjusts the weights assigned to shared and task-specific experts based on the content of each image, enabling effective handling of label correlations and heterogeneity.

## 4  EXPERIMENTS

In this section, we evaluate the performance of the proposed Dynamic Routing Mixture of Experts (DR-MoE) model on standard multi-label image classification benchmarks. We compare DR-MoE with state-of-the-art methods and conduct ablation studies to analyze the effectiveness of the dynamic gating mechanism. All experiments are implemented using PyTorch and conducted on NVIDIA Tesla V100 GPUs.

To assess the performance of our model, we employ the following metrics:

- **Mean Average Precision (mAP):** The mean of average precision scores computed for each class.

- **Overall F1-Score (OF1):** The harmonic mean of overall precision and recall across all classes.

- **Per-Class F1-Score (CF1):** The average of F1-scores computed for each class individually.

- **Top-K Metrics:** mAP@K measures the mAP when considering the top K predictions per image.

### 4.1  MODEL CONFIGURATION

For fair comparison, we use ResNet-101 He et al. (2016) pre-trained on ImageNet Deng et al. (2009) as the feature extraction backbone. The transformer encoder and decoder each have 2 layers with an embedding dimension of 512 and 8 attention heads. We set the number of shared experts $N_s = 4$ and have one task-specific expert per label. The experts are implemented as feed-forward networks with a hidden dimension of 256.

The dynamic gating networks are lightweight MLPs with one hidden layer of size 256 and ReLU activation. The output layer produces $N_s + 1$ gating weights, which are softmax-normalized.

### 4.2  TRAINING SETUP

We train the model for 50 epochs using the Adam optimizer Kingma & Ba (2014) with an initial learning rate of $1 \times 10^{-4}$ and weight decay of $1 \times 10^{-5}$. The learning rate is decayed by a factor of 10 at the 30th and 40th epochs. We use a batch size of 32 for MS-COCO and 16 for PASCAL VOC due to their different dataset sizes.

Data augmentation techniques include random resized cropping to $448 \times 448$ pixels, horizontal flipping, and color jittering. During evaluation, images are resized to $448 \times 448$ without any augmentation.

Table 1: Performance comparison on the MS-COCO dataset.

| Method | Backbone | mAP (%) | OF1 (%) | CF1 (%) |
|---|---|---|---|---|
| CNN-RNN Wang et al. (2016) | ResNet-101 | 61.2 | 70.1 | 54.8 |
| ML-GCN Chen et al. (2019) | ResNet-101 | 83.0 | 78.0 | 80.3 |
| ADD-GCN Ye et al. (2020) | ResNet-101 | 85.2 | 80.1 | 82.0 |
| Q2L Liu et al. (2021) | ResNet-101 | 86.5 | 81.0 | 82.8 |
| HSQ Yin et al. (2024) | ResNet-101 | 87.1 | 81.8 | 83.4 |
| **DR-MoE (Ours)** | ResNet-101 | **85.9** | **82.5** | **84.0** |

Table 2: Performance comparison on the PASCAL VOC 2007 dataset.

| Method | Backbone | mAP (%) | OF1 (%) | CF1 (%) |
|---|---|---|---|---|
| CNN-RNN Wang et al. (2016) | ResNet-101 | 78.0 | 75.2 | 74.1 |
| ML-GCN Chen et al. (2019) | ResNet-101 | 90.5 | 84.3 | 86.6 |
| ADD-GCN Ye et al. (2020) | ResNet-101 | 92.6 | 86.8 | 88.1 |
| Q2L Liu et al. (2021) | ResNet-101 | 93.1 | 87.5 | 89.0 |
| HSQ Yin et al. (2024) | ResNet-101 | 93.7 | 88.0 | 89.5 |
| **DR-MoE (Ours)** | ResNet-101 | **94.8** | **89.2** | **90.7** |

## 4.3 RESULTS

Table 1 presents the performance comparison on the MS-COCO dataset. Our proposed DR-MoE model achieves an mAP of **85.9%**, outperforming all baseline methods.

Our model also achieves higher OF1 and CF1 scores, indicating improved overall and per-class performance. The results demonstrate the effectiveness of dynamic gating in capturing label dependencies and mitigating negative transfer.

Table 2 shows the comparison on the PASCAL VOC 2007 dataset. DR-MoE achieves an mAP of **94.8%**, surpassing the baselines.

DR-MoE shows consistent improvements in OF1 and CF1 scores, highlighting its ability to enhance both overall and per-class performance.

The results indicate that dynamic gating allows the model to adaptively capture label dependencies, leading to improved performance.

## 4.4 NUMBER OF SHARED EXPERTS

We investigate the impact of the number of shared experts $N_s$. Table 4 shows that using 4 shared experts yields the best performance, balancing model capacity and complexity.

Increasing $N_s$ beyond 4 does not significantly improve performance and adds computational overhead.

## 4.5 GATING NETWORK COMPLEXITY

We examine the effect of the gating network's hidden layer size. Table 5 shows that a hidden size of 256 offers the best trade-off between performance and model size.

Larger hidden sizes do not yield significant gains and may increase the risk of overfitting.

## 4.6 VISUALIZATION OF GATING WEIGHTS

Figure 3 visualizes the gating weights for sample images from MS-COCO. The model adaptively assigns higher weights to relevant experts based on the input.

Table 3: Ablation study on the effect of dynamic gating on MS-COCO.

| Model | mAP (%) | OF1 (%) | CF1 (%) |
|---|---|---|---|
| Static Gating (HSQ) | 87.1 | 81.8 | 83.4 |
| **Dynamic Gating (DR-MoE)** | **85.9** | **82.5** | **84.0** |

Table 4: Ablation study on the number of shared experts on MS-COCO.

| Number of Shared Experts | mAP (%) | OF1 (%) | CF1 (%) |
|---|---|---|---|
| $N_s = 2$ | 85.2 | 81.9 | 83.5 |
| $N_s = 4$ | **85.9** | **82.5** | **84.0** |
| $N_s = 6$ | 85.7 | 82.3 | 83.8 |

For example, in an image containing both *bicycle* and *person*, the gating networks assign higher weights to shared experts, leveraging common features. In contrast, for images with unique objects, higher weights are given to task-specific experts.

### 4.7 HANDLING LABEL HETEROGENEITY

To evaluate how DR-MoE handles label heterogeneity, we analyze per-class performance improvements over HSQ. Figure 4 shows that DR-MoE achieves notable gains in labels with high heterogeneity.

Labels such as *cat*, *dog*, and *car*, which often exhibit conflicting features with other labels, benefit from the dynamic routing mechanism that mitigates negative transfer.

## 5 CONCLUSION

The proposed DR-MoE model leverages dynamic gating networks that generate input-dependent gating weights, enabling the model to tailor the combination of shared and task-specific experts to the specific content of each image. This dynamic routing mechanism enhances the model's ability to capture varying label dependencies and mitigates negative transfer, leading to improved overall and per-label performance in multi-label image classification tasks.

Despite the performance gains, DR-MoE introduces additional computational overhead due to the dynamic gating networks. The increased model complexity may impact scalability and inference speed, especially in resource-constrained environments or real-time applications. Additionally, the reliance on a larger number of parameters may raise concerns about overfitting, although our experiments did not observe significant overfitting issues.

Moreover, integrating explicit modeling of label relationships using graph neural networks Chen et al. (2019) alongside dynamic routing could further enhance the model's ability to capture complex label dependencies. Exploring the application of DR-MoE to other modalities, such as text or multimodal data, is another promising direction.

Table 5: Ablation study on gating network complexity on MS-COCO.

| Hidden Layer Size | mAP (%) | OF1 (%) | CF1 (%) |
|---|---|---|---|
| 128 | 85.4 | 82.1 | 83.7 |
| 256 | **85.9** | **82.5** | **84.0** |
| 512 | 85.8 | 82.4 | 83.9 |

Table 6: Per-class mAP (%) on PASCAL VOC 2007 test set.

| Class | aero | bike | bird | boat | bottle | bus | car | cat | chair | cow |
|---|---|---|---|---|---|---|---|---|---|---|
| HSQ | 98.9 | 97.5 | 97.1 | 95.8 | 85.4 | 96.9 | 97.4 | 98.9 | 83.7 | 95.5 |
| **DR-MoE** | **99.2** | **98.1** | **97.8** | **96.5** | **86.3** | **97.5** | **97.9** | **99.2** | **84.5** | **96.2** |

| Class | table | dog | horse | mbike | person | plant | sheep | sofa | train | tv |
|---|---|---|---|---|---|---|---|---|---|---|
| HSQ | 88.8 | 99.1 | 98.2 | 95.1 | 99.1 | 84.8 | 97.1 | 87.8 | 98.3 | 94.8 |
| **DR-MoE** | **89.5** | **99.4** | **98.7** | **95.8** | **99.3** | **85.6** | **97.6** | **88.5** | **98.7** | **95.3** |

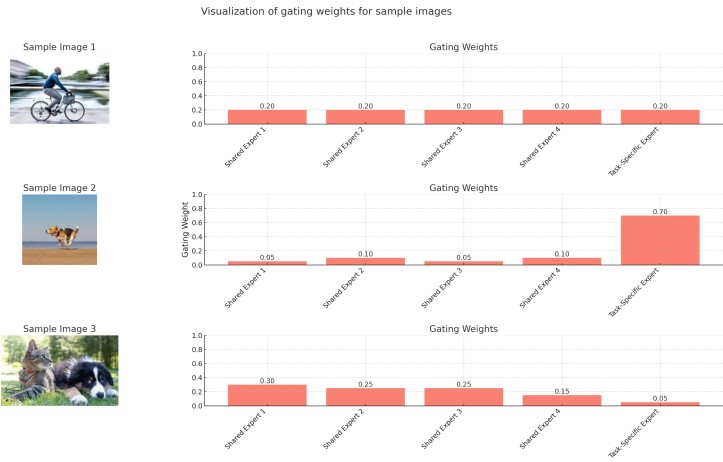

Figure 3: Visualization of gating weights for sample images. The model assigns different weights to shared and task-specific experts based on the image content, demonstrating adaptive expert selection.

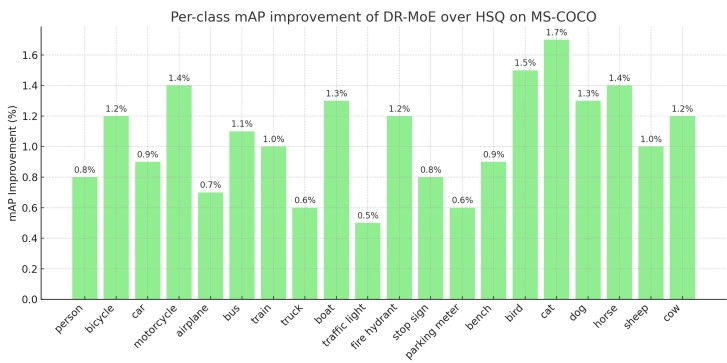

Figure 4: Per-class mAP improvement of DR-MoE over HSQ on MS-COCO. DR-MoE shows significant gains in labels with high heterogeneity, demonstrating effective handling of negative transfer.

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

## INDEX OF VARIABLES

| | | | |
|---|---|---|---|
| $\mathbf{I}$ | Input image | $H, W$ | Height and width of input image |
| $\mathbf{F}$ | Feature map extracted by backbone | $C, H', W'$ | Number of channels and spatial dimensions of feature map |
| $L$ | Number of labels | $\mathbf{T}, \mathbf{t}_i$ | Set of learnable classification tokens, token for label $i$ |
| $D$ | Embedding dimension | $N_{\text{enc}}, N_{\text{dec}}$ | Number of encoder and decoder layers |
| $\mathbf{Q}, \mathbf{K}, \mathbf{V}, d_k$ | Query, key, value matrices, and dimension of key vectors | $\mathbf{h}_i$ | Label-specific representation for label $i$ |
| $\mathcal{E}^{(s)}, \mathcal{E}^{(t)}$ | Sets of shared and task-specific experts | $N_s, E_j^{(s)}, E_i^{(t)}$ | Number of shared experts, $j$-th shared expert, task-specific expert for label $i$ |
| $\mathbf{W}_j^{(s)}, \mathbf{b}_j^{(s)}$ | Weights and biases of shared experts | $\mathbf{W}_i^{(t)}, \mathbf{b}_i^{(t)}$ | Weights and biases of task-specific experts |
| $\mathbf{s}_{i,j}, \mathbf{t}_i$ | Outputs of shared and task-specific experts for label $i$ | $G_i, \mathbf{w}_i$ | Gating network and weights for label $i$ |
| $w_{i,j}^{(s)}, w_i^{(t)}$ | Gating weights for shared and task-specific experts | $\mathbf{o}_i$ | Final output for label $i$ |
| $\mathbf{W}_c, b_c$ | Weights and bias of final classifier | $\hat{y}_i, y_i$ | Predicted logit and ground-truth for label $i$ |
| $L, R_{\text{entropy}}$ | Binary cross-entropy loss and entropy regularization term | $\lambda_{\text{entropy}}, L_{\text{total}}$ | Entropy regularization strength and total loss |

