# OpenReview forum: "Dynamic Routing Mixture of Experts for Enhanced Multi-Label Image Classification"
_ICLR.cc/2025/Conference — Submitted to ICLR 2025_

### Official Review · Reviewer_Ka5g · 2024-10-31

**Soundness:** 1
**Presentation:** 2
**Contribution:** 2
**Rating:** 3
**Confidence:** 4

**Summary:**

This paper introduces the Dynamic Routing Mixture of Experts (DR-MoE) model, a novel architecture for multi-label image classification (MLC). The model utilizes input-dependent dynamic gating within the Mixture of Experts framework, selecting shared and task-specific experts adaptively based on image features. The authors claim that DR-MoE overcomes the limitations of static gating by better capturing label dependencies and mitigating negative transfer. Experiments on the MS-COCO and PASCAL VOC 2007 datasets demonstrate that DR-MoE achieves state-of-the-art performance, outperforming prior methods, with ablation studies showcasing the benefits of dynamic gating in handling label heterogeneity.

**Strengths:**

1.	The proposed approach is simple yet effective, making it easy to follow.
2.	The model demonstrates excellent performance on both the COCO and PASCAL VOC datasets, as supported by experimental results.

**Weaknesses:**

1.	Lack of evidence for statement. The authors argue that learning multiple labels jointly can cause negative transfer due to label heterogeneity. However, this claim lacks empirical evidence or experiments to substantiate it. Intuitively, semantically similar classes often benefit from shared low-level features. Including experimental validation or citation of prior work would strengthen this argument.
2.	Unconvincing motivation. The authors critique HSQ for employing static gating, claiming that the proposed dynamic gating mechanism offers better adaptability (the key motivaition). However, I think HSQ also uses a dynamic gating network to predict routing scores for expert contributions. The example provided (e.g., “an image containing both bicycle and person...” in Line 53-54) does not demonstrate that HSQ cannot handle dynamic label correlations, because HSQ’s predicted routing scores could still allocate task-specific experts as required.
3.	Limited novelty. The primary contribution lies in the input-dependent dynamic gating mechanism, while other components rely on existing, off-the-shelf modules. This limits the novelty of the proposed approach.
4.	Unclear presentation of the Method. The paper suffers from unclear visual representation. For example, Figure 1's overview does not effectively convey the proposed method, and the gating visualizations in Figure 2 are vague, using labels such as “Image 1” and “Image 2” without showing the actual images. PS: A minor issue is the incorrect citation format. The correct format is “\citep” rather than “\cite”.

**Questions:**

The questions are listed in the “weakness” section. I strongly encourage the authors to continue improving their work and explore ways to address these weaknesses.

---

### Official Review · Reviewer_DeXs · 2024-11-01

**Soundness:** 2
**Presentation:** 2
**Contribution:** 1
**Rating:** 3
**Confidence:** 3

**Summary:**

This paper introduces a model—the Dynamic Routing Mixture of Experts (DR-MoE) model, for enhancing multi-label image classification (MLC) tasks. By integrating input-dependent dynamic gating networks into the Mixture-of-Experts (MoE) framework, the model adaptively selects and weights both shared and task-specific experts. It achieves state-of-the-art results on benchmark datasets such as MS-COCO and PASCAL VOC 2007.

**Strengths:**

The paper is easy to follow and details of experiments are carefully described.

**Weaknesses:**

1. The claimed defects in HSQ appear to lack supporting evidence. The proposed method appears to have already been implemented in the baseline model. See Question 1.
2. The novelty and key contribution of this paper are hard to tell.
3. The fig 1 is hard to understand.

**Questions:**

1.	As I gather, the gating networks are not static as the authors claimed in the manuscript. Section 3.3, eq(1, 2) in HSQ explicitly points out that it employs an unique gating network for every label and the gating networks generate weight upon all employed experts with task-specialized feature input. Correct me if I am wrong, this seems to be just the definition of “Dynamic routing” in the submitted manuscript. Gating networks of the proposed model seems to be highly identical to the HSQ’s gating networks.

2.	Author doesn’t mention the resolution of image you used. Are all comparisons under the same image resolution to ensure fair? I also wonder if the same data augmentation methods and other conditions are used in the ablation study on the effect of dynamic gating .
3.	Is the model’s good performance due to the introduction of a large number of parameters? It is recommended to indicate parameters in the experiments.

---

### Official Review · Reviewer_6HwD · 2024-11-01

**Soundness:** 2
**Presentation:** 3
**Contribution:** 2
**Rating:** 3
**Confidence:** 5

**Summary:**

This paper presents a Dynamic Routing Mixture of Experts (DR-MoE) model for the multi-label image classification task, which improves the static gating of the Hybrid Sharing Query (HSQ) by introducing a dynamic gating mechanism that adaptively selects and weighs both shared and task-specific experts based on the input image features. Experiments on the Pascal VOC 2007 and MS-COCO datasets demonstrate the superiority of the proposed DR-MoE model in the multi-label image classification task, and ablation studies on the MS-COCO dataset verify the effectiveness of its key designs.

**Strengths:**

- This paper is generally well organized and written.
- The motivation of the dynamic gating mechanism is reasonable.
- The technical details throughout the paper are well explained and easy to reproduce.

**Weaknesses:**

- The symbol for binary cross-entropy loss and the symbol for the number of labels are both $L$ in Eq.10, which should be differentiated.
- This paper exhibits some experimental results in Table 3 and Table 6. However, the reviewer could not find any explanation about Table 3 and Table 6 in experiment section.
- The dynamic gating network is the core of the DR-MoE model, and merely using the case analysis in Figure 3 to demonstrate its effectiveness in allocating weights for shared experts and task-specific experts is not convincing enough. Authors are encouraged to provide a statistical analysis, e.g., the distribution of weights on these experts across all single-labeled and multi-labeled images in the entire test dataset.
- The authors are encouraged to evaluate the performance of the proposed DR-MoE using additional vision backbones, including TResNet and Vision Transformer.
- The comparison between the DR-MoE model and baseline methods is too weak, and many of the latest multi-label image classification methods have been overlooked. For example, SALGL [1], IDA [2], PAT [3].


[1] Scene-Aware Label Graph Learning for Multi-Label Image Classification. ICCV 2023.

[2] Causality Compensated Attention for Contextual Biased Visual Recognition. ICLR 2023.

[3] Counterfactual Reasoning for Multi-Label Image Classification via Patching-Based Training. ICML 2024.

**Questions:**

- In section 4.3, the author claims that the proposed DR-MoE model outperforms all baseline methods on the MS-COCO dataset. However, the reviewer disagrees with this. According to the Table 1, both Q2L and HSQ exhibit better performance in mAP than DR-MoE. The author should clarify this.
- The HSQ method has multiple task-specific experts, while in DR-MoE, the number of task-specific experts defaults to 1. The reviewer is curious about the impact of the number of task-specific experts on model performance.

---

### Official Review · Reviewer_ZhrH · 2024-11-04

**Soundness:** 2
**Presentation:** 1
**Contribution:** 1
**Rating:** 1
**Confidence:** 5

**Summary:**

The paper proposed a dynamic gating-based multi-task learning for multi-labelled data.

**Strengths:**

The proposal is interesting as it shows a new direction for multi-task learning for multi-labelled data.

**Weaknesses:**

The major weakness of the paper is that the paper is hard to follow. The description of the architecture should be summarized in a diagram.

For example, for professional representation is required, fig 1 is hard to follow.

Another major limitation is that the experimental section is very shallow. as only PASCAL VOC 2007 and MS-COCO are used. There exist many datasets in the literature that should be used, such as Taxonomy, CelebA, CelebMask etc. More experiments on varying segmentation tasks, NLP, retrieval,  tasks etc should be conducted to prove the proposal's effectiveness.

A comparison with many related works in the literature is also missing.

**Questions:**

A proper block digram of the implementation is required.

---

> ### Comment · Reviewer_ZhrH · 2024-11-26
> **No author respose**
>
> No author response is been posted.

---

### Meta-Review · Area_Chair_KqvJ · 2024-12-08

**Metareview:**

At the initial review stage, all the reviewers have strong negative opinions.

The concerns are mainly centered around writing and clarity (e.g., Fig1), weak experimental results (only PASCAL VOC 2007 and MS-COCO), lack of discussion for related works (e.g., SALGL, IDA, PAT as Reviewer 6HwD pointed out).

As the authors did not provide a rebuttal and the AC agrees with the initial reviews by the reviewers, the AC recommends the rejection of this paper.

**Additional Comments On Reviewer Discussion:**

No rebuttal has been provided.

---

### Decision · Program_Chairs · 2025-01-22

Reject